# Neryl acetate, the major component of Corsican *Helichrysum italicum* essential oil, mediates its biological activities on skin barrier

Géraldine Lemaire[1]*, Malvina Olivero[1], Virginie Rouquet[1], Alain Moga[2], Aurélie Pagnon[3], Valérie Cenizo[1], Pascal Portes[1]

1 Laboratoires M&L SA–Groupe L'Occitane, Manosque, France, 2 QIMA Life Sciences–Synelvia, Labège, France, 3 Novotec, BRON, France

* geraldine.lemaire@loccitane.com

**Data Availability Statement:** All relevant data are within the paper and its Supporting Information files

## Abstract

Corsican *Helichrysum italicum* essential oil (HIEO) is characterized by high concentrations of neryl acetate, and we previously demonstrated that Corsican HIEO increases the expression of genes that are part of the differentiation complex (*involucrin*, *small proline rich proteins*, *late cornified envelope*, *S100 protein family*). The biological activities of HIEO and neryl acetate (NA) were compared to identify how NA contributes to HIEO activity on human skin. NA, as a part component of HIEO, was tested on skin explant models for 24 hours and 5 days in comparison with HIEO. We analyzed the biological regulations in the skin explant by transcriptomic analysis, skin barrier protein immunofluorescence, lipid staining and ceramide analysis by liquid chromatography-mass spectrometry. Transcriptomic analysis revealed that 41.5% of HIEO-modulated genes were also regulated by NA and a selected panel of genes were confirmed by qquantitative reverse transcription PCR analysis. Those genes are involved in epidermal differentiation, skin barrier formation and ceramide synthesis. Involucrin (IVL), involved in formation of the cornified envelope (CE), was upregulated at both gene and protein levels after 24 hours and 5 days respectively. After 5 days of treatment, total lipids and ceramides were also increased. Our results demonstrate that NA mediates a large part of Corsican HIEO activity on skin barrier formation.

## Introduction

Helichrysum italicum (Roth) G. Don subsp. *italicum* (H. italicum italicum) from the Asteraceae family grows on dry, rocky or sandy ground around the Mediterranean Basin and is commonly known as the everlasting plant. The stems are woody at the base and can reach 30–70 cm in height [1]. It is a small aromatic shrub with yellow flowers and blossoms in June [2]. HIEO is obtained from the hydrodistillation of aerial parts [3] and its composition is known to depend on the geographical area of collection [2,4]. Indeed, the chemical composition of the HIEOs collected in Sicily and Corsica analyzed by Schipilliti *et al.* [5] shows that Sicilian oils

**Funding:** This work was supported by Laboratoires M&L SA - Groupe L'Occitane. The funder provided support in the form of salaries for authors [G.L, M. O, V.R, V.C, P.P] and research materials, but did not have any additional role in study design, data collection and analysis, decision to publish, or preparation of the manuscript. A.M is employed by QIMA – Synelvia and A.P is employed by Novotec. The specific roles of each author are articulated in the 'author contributions' section.

**Competing interests:** The financial support from Laboratoires M&L - Groupe L'Occitane does not alter the authors' adherence to all PLOS ONE policies on sharing data and materials and there are no restrictions on sharing of data and/or materials.

are rich in α- and β-selinene, rosifoliol and aromadendrene, whereas Corsican oils are rich in NA, α-pinene and γ-curcumene. The highest NA content found in Corsican HIEO from plants collected in the flowering stage is correlated with relatively weak soil acidity and low percentages of clay, fine sand and coarse silt and with high percentages of coarse sand and fine silt [6].

H. italicum is traditionally used in respiratory, digestive and dermatological disorders (cough, alopecia, inflammation) [1]. Other therapeutic applications, including antimicrobial activity, wound healing, digestive disorders and analgesic uses, have been reported [1].

Essential oil from H. italicum italicum exhibits an inhibitory effect on gram-positive bacteria growth, inhibits the growth of *Staphylococcus aureus* in a dose-dependent manner [7] and is also effective against *Candida albicans* [8], mainly because of its terpenoid fraction. Recently, Fraternale et al. [9] demonstrated the *in vitro* anti-collagenase and anti-elastase activities of Italian HIEO containing NA (15.75%), α-pinene (8.21%) and limonene (4.55%). They hypothesized that limonene and *a*-pinene may be responsible for these activities.

NA is an acetate ester resulting from formal condensation of the hydroxy group of nerol with the carboxy group of acetic acid. This volatile metabolite contributes to the plant's fragrance. Corsican HIEO is characterized by a high level of NA (33.7–38.9%) [2,10]. No biological activity of NA is described but geranyl acetate, an NA isomer, has antifungal activity [11] and induces potent anticancer effects in Colo-205 colon cancer cells by inducing apoptosis, DNA damage and cell cycle arrest [12].

The *stratum corneum* (SC) provides the body's main barrier to the environment and is key to maintaining optimal cutaneous hydration. The barrier structure of the *stratum corneum* in human skin has four major components: the corneocyte, surrounded by the cornified envelope (CE) and the cornified lipid envelope (CLE), which are embedded in the intercellular lipid layers [13]. The formation of the CE is characterized by the expression of a set of genes, such as *loricrin* (*LOR*), *IVL*, *transglutaminase* (*TGM1*) and *filaggrin* (*FLG*). *IVL* is expressed at an early stage of keratinocyte differentiation and promotes CE formation [14]. It provides mechanical strength by a scaffold to which other proteins then become to cross-link to IVL. In the CE structure, IVL is adjacent to the cell membrane and forms the exterior surface of the CE. Keratin (KRT)/FLG and their degradation products (natural moisturizing factor, NMF) fill the intracellular matrix of corneocytes [15]. The CLE is mainly composed of ultra-long-chain (ULC) ceramide and ULC fatty acids. As the bond between the outer surface of the CE and the inner surface of the CLE, glutamyl residues of IVL and other CE precursor proteins covalently bind to -hydroxy residues of -hydroxy ULC-ceramides and -hydroxy ULC fatty acids; this crosslinking is catalyzed by TGM1 [16]. LOR, expressed in the granular layer during cornification, is the main component of the epidermal CE and represents 70–85% of the total protein mass of the cornified layer. LOR functions as the main reinforcement protein for the CE on its intracellular face [17].

The most important components of the intercellular lipid layers are ceramides (50%), cholesterol (25%) and free fatty acids (10–20%) [18]. The lipids are organized in stacked bilayer structures with alternating periodicities of 13 nm (long periodicity phase) and 6 nm (short periodicity phase) [19]. Disturbance in the lipid composition and/or organization is related to changes in the barrier properties and skin diseases (atopic dermatitis, psoriasis. . .) [20]. Lipids adhere to the SC via lamellar bodies that release SC lipid precursors and hydrolytic enzymes by exocytosis into the intercellular space [18].

The free fatty acids in the SC are predominantly straight chained and dominated by saturated long-chain fatty acids, mainly C24:0 and C26:0 [21]. Cholesterol is the major sterol in the SC, but cholesterol sulfate is another important skin lipid involved in the keratinocyte differentiation and desquamation process. It is generated by sulfotransferase family 2B member 1 (SULT2B1) from cholesterol and is located mostly in the upper part of the epidermis, with low concentrations in the SC [22].

At least thirteen different subtypes of ceramides are produced in the epidermis, which are composed of long chain sphingoid bases linked to fatty acid groups (FA) via an amide bond. The *de novo* synthesis of ceramides involves serine palmitoyl-CoA transferase (SPTLC), ceramide synthase (CERS) and glucosylceramide synthase (UGCG). The four types of sphingoid bases represented in ceramides are dihydrosphingosine (DS), sphingosine (S), phytospingosine (P) and 4-hydroxysphingosine (H) [23]. The FAs in ceramides are classified into non-hydroxylated FA (N), α-hydroxylated FA (A), ω-hydroxy FA (O) and esterified ω-hydroxy FA (EO). Among ceramides, the most abundant of these subclasses in the *stratum corneum* are NP, NH, AP and AH mainly with fatty acid chain lengths C24 and C26 and with small contributions from C16 and C18, the latter being the most abundant chain length of sphingoid base [23]. Ceramides constitute a hydrophobic extracellular lipid matrix, indispensable for permeability barrier function, and act as active second messengers, regulating keratinocyte proliferation and differentiation, enhancing proinflammatory cytokine production and modulating immune response [24].

During aging, the skin barrier function is reduced and the composition of the CE changes dramatically due to altered gene coding expression patterns for major components of the CE. There is also an overall reduction in *stratum corneum* lipids and a disturbance in cholesterol and fatty acid synthesis. Ceramide [EOS] is significantly reduced in seniors ($> 50$ years) compared to younger individuals (20–40 years). Moreover, the degree of fatty acid chain saturation of ceramide [EOS], which has marked effects on lamellar and lateral lipid organization, decreases in autumn and winter, partially accounting for worse barrier function. Consequently, there is also evidence of altered permeability barrier to chemical substances and increased trans epidermal water flux in aged skin [25].

We previously demonstrated that 24-hour treatment with Corsican HIEO upregulates genes involved in the epidermal differentiation complex: *IVL*, *small proline rich proteins (SPRRs)*, *late cornified envelopes (LCEs)*, *S100 calcium binding proteins (S100)* in skin explants [26]. NA as a part component of HIEO was tested on skin explants and its activity compared to that of the HIEO containing it. Transcriptomic analysis revealed that 41.5% of HIEO-modulated genes are also regulated by NA; those genes are related to skin barrier formation (keratinocyte differentiation, epidermal differentiation complex and junctions) and ceramide synthesis. Skin barrier formation is a complex process involving the coordinated expression of structural proteins of the CE such as IVL and enzymes responsible for the production of epidermal lipids and ceramides. We focused on IVL, which is upregulated at both gene and protein level by NA and HIEO treatments. Total lipids and ceramides were also increased, and this was correlated with the expression of genes involved in lipid and ceramide synthesis pathways. Those results demonstrated in our experimental conditions that NA mediates the skin barrier formation induced by Corsican HIEO.

## Materials and methods

### Plant material and oil distillation

Aerial parts of H. italicum italicum were collected at flowering time in July from a crop cultivation located on the Corsican coast. Fresh aerial parts were hydrodistilled for five hours using a Clevenger-type apparatus.

### Gas chromatography–mass spectrometry analyses

Gas chromatography analysis was performed on a gas chromatograph equipped with a flame ionization detector, using a TR-Wax MS-fused silica capillary column (Thermo Fisher Scientific, 60 m x 0.25 mm i.d.; film thickness 0.25 μm). Chromatographic conditions were as

follows: hydrogen as carrier gas at 0.7 mL/min and injector and detector temperatures at 250˚C each. Oven temperature was isothermal at 60˚C for 1 min, then increased to 240˚C at a rate of 2˚C/min and held isothermal for 23 mins. The volume injected was 1 μL with a split ratio of 200:1. Identification was performed with standards and/or NIST database. In this study, only retention time and comparison with standards is used and GC-MS analysis before GC-FID analysis confirm the compound identification (Table 1).

## Human skin biopsy collection and treatment

Full-thickness biopsies of abdominal skin, collected during cosmetic surgery, were purchased from suppliers accredited by the French Ministry of Research: Alphenyx (Marseille, France) and DermoBioTec (Lyon, France). The tissue collection used in this study included 3 biopsies of abdominal skin from 33-, 34- and 35-year-old women for histology, microarray analysis, IVL and lipid staining, and 3 from 36-, 47- and 49-year-old women for ceramide analysis.

HIEO and NA were solubilized in DMSO at 20% and 6%, respectively. DMSO never exceeded 0.5% in culture medium and was considered as the control.

Treatment consisted of DMSO at 0.5% as control, HIEO at 0.1% or NA at 0.03% in the culture medium at day 0 and day 2.

Quadruplicates from each donor were collected 24 hours after treatment for genomic expression analysis and at day 5 for morphological evaluation by histology, IVL detection, lipids and ceramides analysis.

## Histological analysis

Explants at day 5 were immediately fixed in neutral buffered formalin 10% (Sigma-Aldrich, Saint-Quentin-Fallavier, France) for 24 hours and embedded in paraffin. Paraffin-embedded formalin-fixed samples were then cut into 5-μm sections. After dewaxing and rehydration, sections were stained with Masson's trichrome.

**Table 1. Chemical composition of HIEO from Corsica.**

| N˚ | Tr (min)[a] | Compound | Percentage |
|---|---|---|---|
| 1 | 8.50 | α-pinene | 2.02 |
| 2 | 10.98 | β-pinene | 0.68 |
| 3 | 14.60 | Limonene | 4.83 |
| 4 | 14.99 | 1,8-cineol | 0.82 |
| 5 | 30.68 | α-cedrene | 0.63 |
| 6 | 33.30 | Linalool + italicene | 4.29 |
| 7 | 36.30 | β-caryophyllene | 1.57 |
| 8 | 41.80 | γ-curcumene | 12.31 |
| 9 | 43.77 | Neryl acetate | 32.80 |
| 10 | 46.08 | α-curcumene | 2.59 |
| 11 | 46.95 | Neryl propionate | 4.66 |
| 12 | 47.22 | Nerol | 2.01 |
| 13 | 51.94 | Italidiones 1 and 2 | 10.34 |
| 15 | 63.21 | β-eudesmol | 3.57 |

[a]: Retention time (min) of corresponding constituent obtained by GC–FID, Percentage = relative abundance.

## IVL immunofluorescence, image acquisition and analysis

Labeling was performed on air-dried 5-μm paraffin sections and incubated with anti-IVL rabbit IgG (6011008, Novocastra). Secondary Alexa Fluor® 488 goat anti-rabbit IgG antibody (Invitrogen, Asnières-sur-Seine, France) was incubated for one hour at room temperature. Nuclear acid counterstaining using DAPI (4′,6-diamidino-2-phenylindole) was carried out routinely. As a negative control, primary antibody was replaced by the corresponding control isotype.

Immunofluorescence specimens were visualized using a DMLB Fluorescence Microscope (Leica), and images were captured using a DFC420C digital camera (Leica). Five representative images were captured for each four replicates by condition and eight-bit images were saved. They were processed and analyzed using ImageJ software for microscopy (http://www.macbiophotonics.ca/imagej/, Research Service Branch, US National Institutes of Health, United States). The surface area of involucrin immunostainings was measured and the mean of five representative images per replicate was calculated.

## Lipid staining

Explants at day 5 were immediately fixed in OCT compound and frozen at -80˚C. OCT-fixed samples were then cut into 5-μm sections and after fixation with propylene glycol, skin cryosections were incubated in Oil Red O Solution (Sigma-Aldrich, Saint-Quentin-en-Yvelines, France) for five minutes at room temperature, incubated in Mayer's hematoxylin for one minute, and mounted with coverslip and aqueous mounting medium (Faramount, Dako).

## Transcriptome analysis

**RNA extraction.** For each sample, a small piece of skin (90–100 mg) was disrupted and homogenized using Omni Tissue Homogenizer into TRIzol® reagent. The upper phase was transferred to RNeasy spin columns and total RNA was extracted using mini-RNeasy kits (Qiagen, Courtabœuf, France) according to the manufacturer's instructions, with the addition of the DNase digestion step. RNA quality/integrity and concentration were assessed using an Agilent 2100 Bioanalyzer (Agilent Technologies, Les Ulis, France) and NanoDrop™ Spectrophotometer (Thermo Fisher Scientific, Asnières-sur-Seine, France), respectively.

**Preparation and hybridization of probes.** The total RNA (100 ng) of each sample was reverse transcribed, amplified and labeled with Cyanine 3 (Cy3) as instructed by the manufacturer of the One-Color Agilent Low Input Quick Amp Labeling Kit (Agilent Technologies, Les Ulis, France). Cy3-labeled cRNA was hybridized onto Agilent Whole Human Genome Oligo 8X60K V2 Arrays (SurePrint G3 Human Gene Expression 8x60K v2 Microarray Kit, G4851B; 50,599 probes) using reagents supplied in the Agilent Hybridization kit (One-Color Microarray-based Gene Expression Analysis Protocol). The slides were scanned with the Agilent SureScan Microarray Scanner System. The one-color microarray images were extracted with Feature Extraction software (v12.0.0.7), which performs background subtractions and generates a quality control report.

**Microarray analysis.** Raw data produced from microarrays were imported into Gene-Spring GX13.0 software (Agilent) to determine the differentially expressed genes between treated and control samples. The raw data were normalized and filtered by flag using Gene-Spring GX13.0 software. The normalization included log2 transformation, per chip normalization to 75% quantile and dropped per gene normalization to median. Flag filtering included genes that were at least detected in 100 percent of the samples in either condition. In addition, GeneSpring GX13.0 software was used to filter a set of HIEO- or NA-responsive genes for which the expression levels were significantly modified (unpaired moderated t-test, $p < 0.05$),

with an average fold change $\geq 2$ or $\leq -2$ compared to the control group (DMSO 0.5%). This result was adjusted for multiple testing by the false discovery rate (FDR) with the Benjamini–Hochberg procedure using a threshold of 0.05. HIEO- or NA-responsive genes were investigated by Ingenuity Pathway Analysis (IPA, Qiagen, Redwood City, CA).

**Bioinformatic analysis.** Bioinformatic analysis was performed to identify HIEO- or NA-responsive genes. Briefly, outcomes from microarray analysis were first uploaded into Qiagen's IPA system for core analysis and then overlaid with the global molecular network in the Ingenuity Pathway Knowledge Base (IPKB). IPA was performed to identify canonical pathways, diseases and functions, pinpoint gene networks that are most significant to microarray outcomes and categorize differentially expressed genes in specific functions. Heatmap and hierarchical cluster analysis were used to demonstrate the expression patterns of these differentially expressed genes. z-score analysis, used as a statistical measure of the match between expected relationship direction and observed gene expression of the uploaded dataset. Positive and negative **z** scores indicate up-regulated and down-regulated pathways, respectively. In line with IPA cut-off values, **z** scores of $\geq 2 \cdot 0$ or $\leq -2 \cdot 0$ were considered significant.

## QRT-PCR

The reverse transcription reaction was performed on 0.4 μg of RNA template and cDNA was synthesized using an AffinityScript QPCR cDNA Synthesis Kit as instructed by the manufacturer (Agilent Technologies, Les Ulis, France). Quantitative PCR was performed using optimized TaqMan® gene expression assays (Applied Biosystems) (Table 2) with Brilliant III Ultra-Fast QPCR Master Mix (Agilent Technologies, Les Ulis, France) on the MX3005P Stratagene instrument. Each PCR reaction was done in duplicate. Results were normalized to the expression of the housekeeping genes POLR2A, GAPDH and TBP. The reaction profile consisted of an initial denaturation at 95˚C for 3 minutes followed by 45 cycles of PCR at 95˚C for 20 seconds (denaturation), 60˚C for 20 seconds (annealing) and 72˚C for 10 seconds (extension). qPCR data was analyzed using qBase software (v 2.3) for the management and automated analysis of qPCR data. Expression of target genes was quantified based on the ΔCq method modified to consider gene-specific amplification efficiencies and multiple reference genes.

**Table 2. Primers used for TaqMan real-time PCR.**

| Gene name | TaqMan Probe | Assay ID |
|---|---|---|
| Aquaporine 3 | AQP3 | Hs00185020_m1 |
| ATP binding cassette subfamily member 12 | ABCA12 | Hs00292421_m1 |
| Ceramide synthase 3 | CERS3 | Hs00698859_m1 |
| ELOVL fatty acid elongase 4 | ELOVL4 | Hs00224122_m1 |
| Involucrin | IVL | Hs00846307_s1 |
| Sulfotransferase family 2B member 1 | SULT2B1 | Hs00190268_m1 |
| Transglutaminase 1 | TGM1 | Hs00165929_m1 |
| UDP-glucose ceramide glycosyltransferase | UGCG | Hs00916612_m1 |
| Glyceraldehyde 3 phosphate dehydrogenase | GAPDH | Hs99999905_m1 |
| Hypoxanthine phosphoribosyltransferase 1 | HPRT1 | Hs99999909_m1 |
| RNA polymerase II subunit A | POLR2A | Hs00172187_m1 |
| TATA-box binding protein | TBP | Hs00427620_m1 |

### Ceramide extraction and analysis

Ceramide content in the epidermis were extracted from tissues by a dual liquid–solid extraction method. The lipid solution was then completely dried under $N_2$ at 30˚C. Residues were re-suspended in 50 μL of a chloroform/methanol mixture. Ceramides were analyzed by a LC/MS Ultimate 3000 liquid chromatography system coupled to a MSQ Plus single quadrupole mass spectrometer (Thermo Fisher Scientific, Sunnyvale, CA).

In this LC/MS system, two mobile phases—M1 (methanol/water (50:50, v/v) and M2 (methanol/isopropanol (80:20, v/v))—were eluted at a flow rate of 0.2 mL/min. The mobile phases were programmed consecutively as follows: a linear gradient of 100–0% M1 between 0 and 20 mins, an isocratic elution of 0% M1 for 30 mins, and then an isocratic elution of 100% M1 for 10 mins. The injection volume was 20 μL and column temperature was maintained at 40˚C. For MS detection, atmospheric pressure chemical ionization was used as the ion source.

## Results

### NA drives 41.5% of HIEO gene regulation

We performed a transcriptome analysis using Agilent Whole Human Genome Oligo Microarrays 8x60K V2 on skin explant from three age-related donors (34 ± 1 years old) treated with the vehicle alone (DMSO 0.5%), HIEO (0.1%) or NA (0.03%), representing the percentage of NA found in HIEO (30%), and experiments were carried out independently in quadruplicate. A two-fold change threshold together with a *p*-value of 0.05 was used to identify differentially expressed genes. Fig 1 depicts the numbers and overlap in differentially expressed genes ($\geq$ 2.0-fold or $\leq$ -2.0-fold, unpaired t-test *p*<0.05) in skin explant following HIEO or NA exposure. The Venn diagram illustrated that HIEO and NA have 1,008 expressed genes (S1 Table) in common, indicating that 41.5% expressed genes regulated by HIEO treatment were also regulated by NA. The 129 and 1,421 genes identified as unique to NA and HIEO treatment are listed in S2 and S3 Tables, respectively.

A total of 59 enriched GO terms for biological processes, molecular functions, and cellular components (p value < 0.05) are shown in S4 Table, related to 1,008 genes common to NA

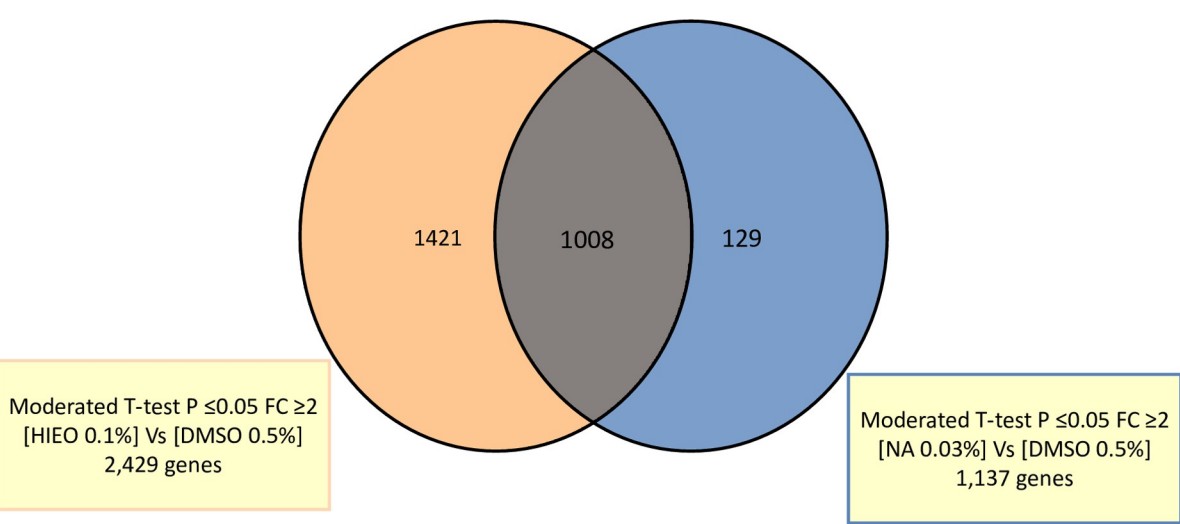

**Fig 1. HIEO and NA share 1,008 regulated genes.** Venn diagram showing the number of similarly and differentially regulated genes between HIEO and NA treatment. Numbers in no overlapping sections represent genes unique to HIEO or NA conditions, while numbers in overlapping areas represent genes shared by HIEO or NA conditions.

and HIEO. The most highly enriched biological processes are related to type I interferon, xenobiotics, the metabolic process of amino acids modified at the cellular level, the process of carboxylic acid biosynthesis, skin and epidermis development, and lipid localization. As shown in Fig 2, the addition of genes uniquely regulated by NA or HIEO enriched some pathways such as skin and epidermis development, lipid localization, cornified envelope for both treatments.

The common 1,008 genes between NA and HIEO were scrutinized with IPA (Qiagen, Redwood City, CA, USA), revealed notably activation z-score of the pathway identified as "skin differentiation" to be equal to 2.3, with 30 genes highlighted (Fig 3). All these results strongly suggest that NA and HIEO could induce skin differentiation. Fig 3, which details this pathway, shows that the 30 genes involved in this biological function are related to skin barrier formation, keratinocyte differentiation and ceramide synthesis. Our study therefore focused on these pathways by identifying other genes in addition to these 30 genes that enter into their regulation.

To validate genes identified by microarrays, we selected eight upregulated genes (*ABCA12*, *AQP3*, *CERS3*, *ELOVL4*, *IVL*, *SULT2B1*, *TGM1 and UGCG*) for qPCR confirmation. Data were presented as fold changes in gene expression normalized to four housekeeping genes (*GAPDH*, *HPTR1*, *POLR2A*, *TBP*) and relative to the DMSO control sample. qPCR analysis (independent RNA extractions from biological replicates used for microarray analysis) confirmed the direction of change detected by microarray analysis (Fig 4). This correlation indicated the reliability of microarray results. The primers used in the qPCR analysis are shown in Table 2.

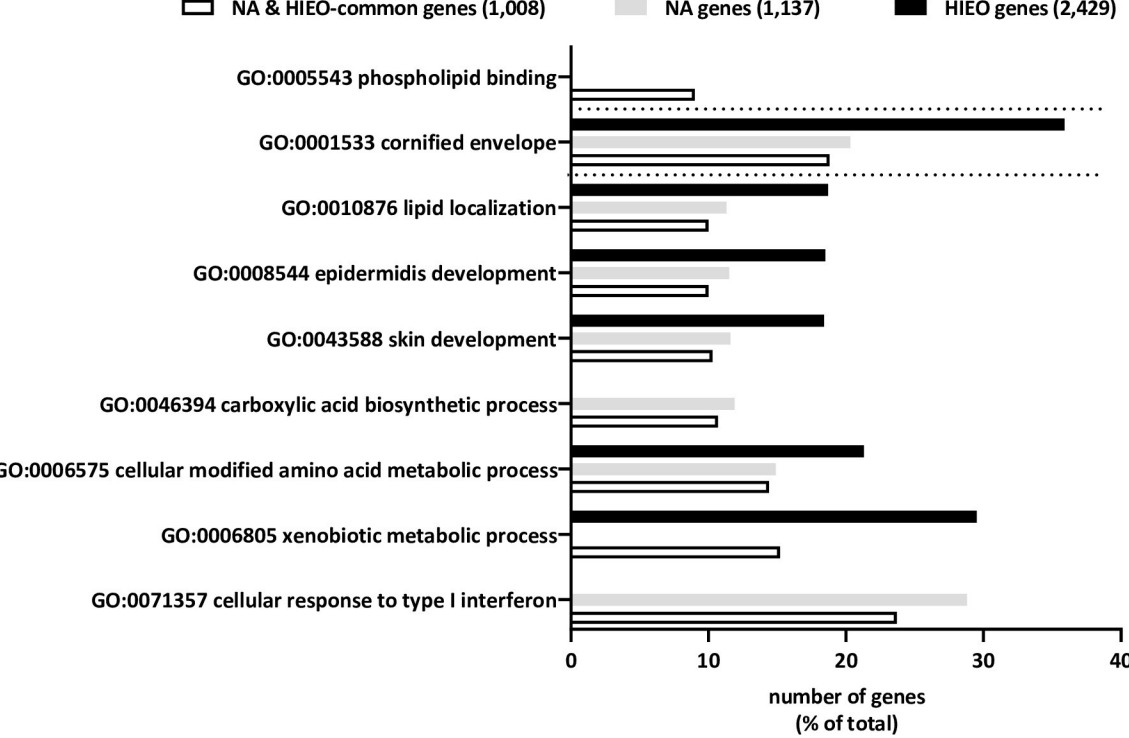

**Fig 2. The top enriched GO terms.** The GO terms were obtained from pathway enrichment analysis of differentially expressed genes (fold changes ≥2 or ≤-2, Benjamini-Yekutieli corrected p-values to take care of multiple GO term testing ≤ 0.05). Those significantly associated (P ≤ 0.05) with the gene list are plotted with the number of genes (as a percentage of the list total) represented by bars, modulated by each treatment. Some terms were excluded as being redundant or having wide meaning (S4 Table).

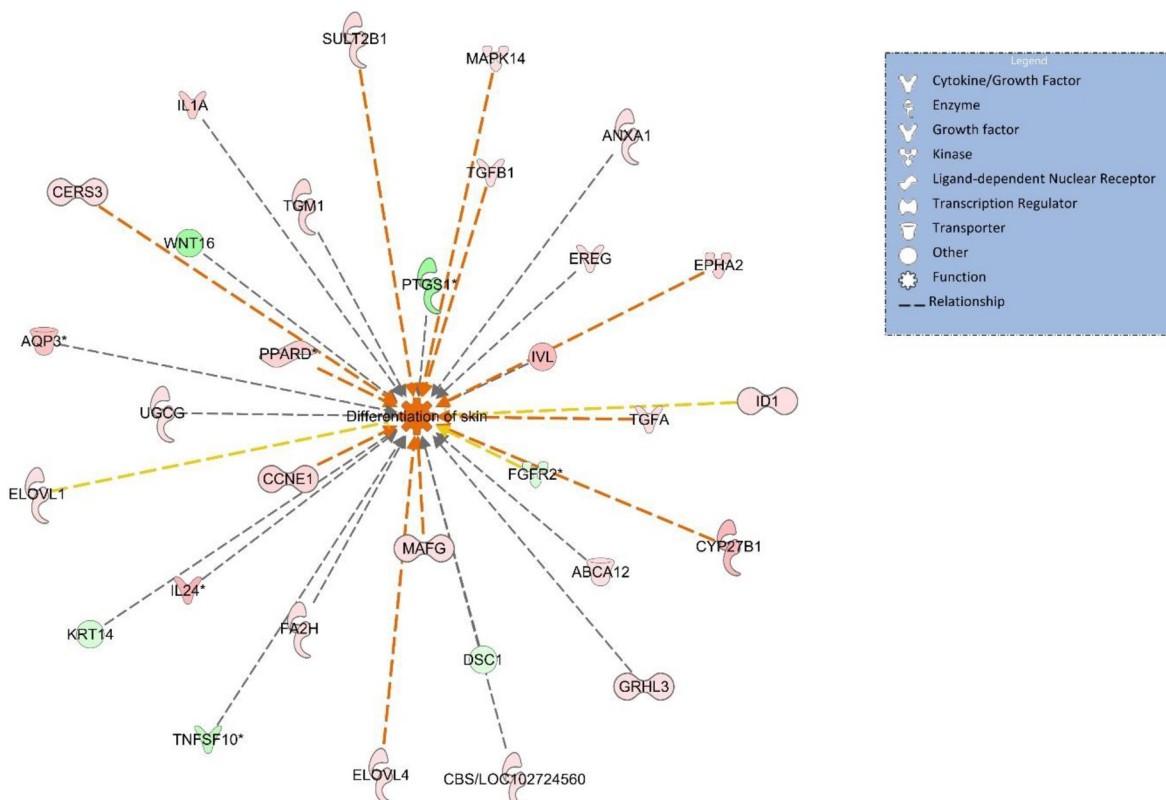

**Fig 3. Biological function analysis using IPA predicts activation of skin differentiation from differentially expressed genes shared by NA and HIEO conditions (*p*-value: <0.05, Z-score: 2.3).** Z-score indicates the predicted activation (positive value) or inhibition (negative value) of the biological function and p-value was calculated by right-tailed Fisher's exact test. Figure represents genes associated with the biological function "skin differentiation" which are altered in the uploaded dataset. Genes that are upregulated are displayed within red nodes and those downregulated are displayed within green nodes. The intensity of color in a node indicates the degree of up-(red) or down-(green) regulation. An orange line indicates predicted upregulation, whereas a blue line indicates predicted downregulation. A yellow line indicates expression being contradictory to the prediction. A gray line indicates that direction of change is not predicted. Solid or broken edges indicate direct or indirect relationships, respectively.

## NA increases formation of the skin barrier

The epidermal differentiation in the presence of NA or HIEO is favored by the upregulation or downregulation of certain genes. They are either involved in mechanisms leading the differentiation/proliferation of keratinocytes or are markers of a differentiated state. As shown in Fig 5A, *cystathione beta-synthase (CBS/CBSL)*, *cyclin E (CCNE1)*, *grainyhead like transcription factor 3 (GRHL3)*, *EPH receptor A2 (EPHA2)*, *cathepsin L (CTSL)*, *peroxisome proliferator activated receptor delta (PPARδ)* and *RAS related 2 (RRAS2)*, which are involved in the differentiation process, were upregulated while *prostaglandin-endoperoxide synthase 1 (PTGS1)* and *POU class 3 homeobox 1 (POU3F1)*, two differentiation repressors, were downregulated. *Vav guanine nucleotide exchange factor 3 (VAV3)* and *Wnt family member 16 (WNT16)*, which regulate proliferation, decreased (Fig 5A). S1 Fig shows that HIEO treatment specifically down-regulated the expression of genes that regulate keratinocytes proliferation as *tumor protein p63 (TP63)*, *E2F transcription factor 1 (E2F1)*, *marker of proliferation Ki-67 (MKI67)*, *forkhead box M1 (FOXM1)* and inversely the expression of *PPARF061*, *E74 like ETS transcription factor 3 (ELF3)*, *RUNX family transcription factor 1 (RUNX1)* but *GATA binding protein 3 (GATA3)* involved in differentiation process, were increased in addition to those

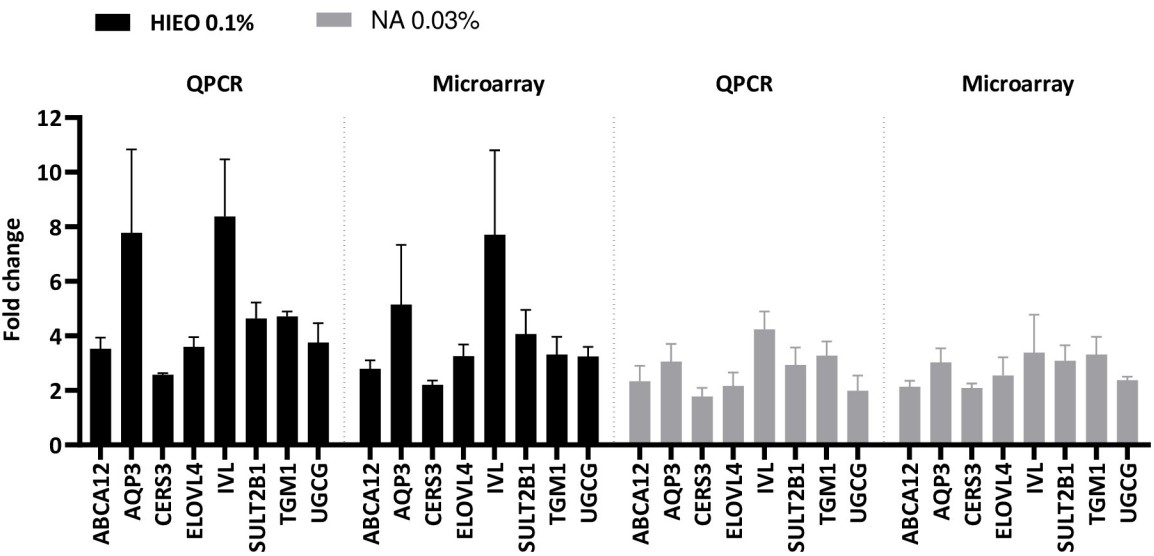

**Fig 4. QRT-PCR confirms HIEO and NA changes identified in microarray expression data.** Results indicate relative gene expressions of HIEO- (0.1%) or NA-treated skin (0.03%) compared to the control DMSO (0.5%). QRT-PCR results for a subset of genes identified as skin barrier (*AQP3*, *IVL*, *TGM1*), ceramide biosynthesis (*ABCA12*, *CERS3*, *ELOVL4*, *UGCG*), and cholesterol metabolism (*SULT2B1*) validated the microarray expression data for skin explants. Data, from 3 individuals in quadruplicate for each donor, are expressed as the gene of interest corrected by the geometric mean of POLR2A, GAPDH and TBP and are the mean ± standard error.

described common to HIEO and NA treatment. Only *ERBB receptor feedback inhibitor 1* (*ERRFI1*) that regulates keratinocytes differentiation, was upregulated by NA treatment specially.

In parallel, *keratin 14* (*KRT14*) expressed in the basal layer of the epidermis and its partner *keratin 5* (*KRT5*) were downregulated by both treatments. *Keratin 10* (*KRT10*) and *keratin 1* (*KRT1)* expressed in the differentiated suprabasal cells were downregulated and *keratin 2* (*KRT2*) was also downregulated. Late differentiation markers such as *IVL*, was upregulated by NA and HIEO treatment. *SPRR3* and *cornulin* (*CRNN)*, other genes of the epidermal differentiation complex (EDC), and *AQP3* and *TGM1*, involved in terminal differentiation of the epidermis, were upregulated while *LCE2A*, another EDC gene, was downregulated by NA and HIEO treatment (Fig 5B). S2 Fig shows that HIEO treatment specifically up- or down-regulated the expression of horny cell layer protein-coding genes in addition to those described common to HIEO and NA treatment.

Intercellular junctions were also upregulated with genes related to gap junctions: *gap junction protein beta 3 (GJB3)* and *pannexin 2* (*PANX2*), tight junctions: *claudin 7* (*CLDN7*) and desmosomes: *desmoglein 3* (*DSG3)* and *plakophilin 2* (*PKP2*) (Fig 5C).

The growth factors upregulated in our study included cell proliferation and/or migration regulators such as members of the EGF family: *heparin binding EGF like growth factor* (*HBEGF*), *epiregulin* (*EREG*), *amphiregulin* (*AREG*), *epithelial mitogen* (*EPGN*) but two of their receptors, *fibroblast growth factor receptor 2/3* (*FGFR2/FGFR3)*, were downregulated by NA and HIEO treatment (Fig 5D).

Fig 6A compares the morphology of the explant at day 5 between control solvent (DMSO 0.5%), 0.1% HIEO or 0.03% NA. There are no discernible tissue structure alterations following treatment.

Immunohistochemical staining confirmed an increased expression of IVL protein in HIEO- and NA-treated skin. IVL staining is increased in spinous layer following treatment with HIEO or NA (Fig 6B). Protein quantification by image analysis (Fig 6C) confirmed a

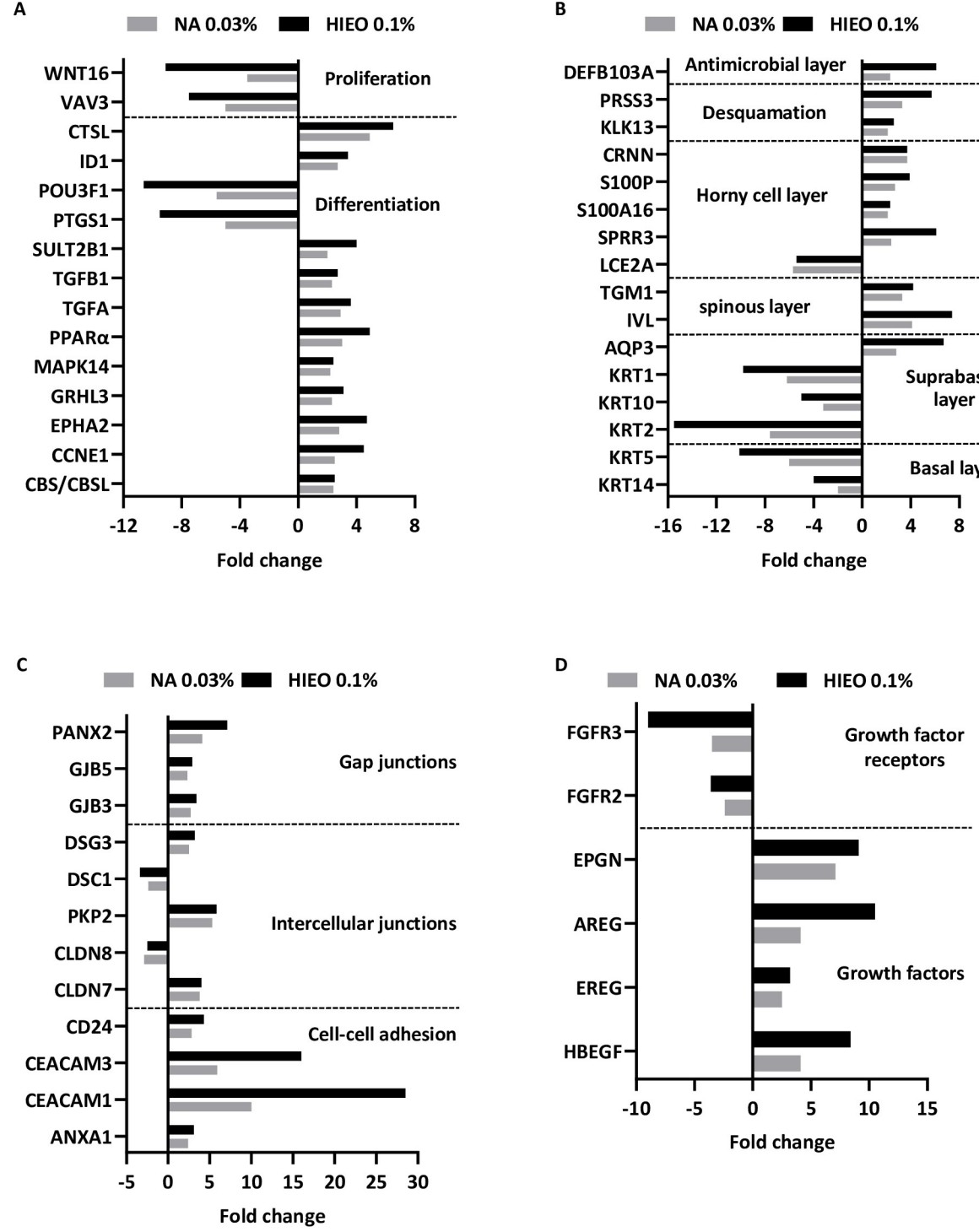

**Fig 5. NA increased the expression of genes involved in barrier formation and its reinforcement.** Transcriptome analysis was performed with microarray using skin explant treated with NA (0.03%) or HIEO (0.1%) and a control group, DMSO (0.5%). Mean fold-change induced by HIEO and NA in quadruplicate explant for each donor from three distinct donors after 24-hour treatment are shown. The 1008 genes in common to HIEO and NA were analyzed with IPA software and an enrichment analysis of certain pathways allowed the identification of differentially expressed genes (fold changes ≥ 2 or ≤ -2, FDR p-value < 0.05) involved in **A:** The differentiation and proliferation of keratinocytes. **B:** In the structure of the epidermis. **C:** In cellular junctions. **D:** The expression of genes encoding for growth factors and their receptors.

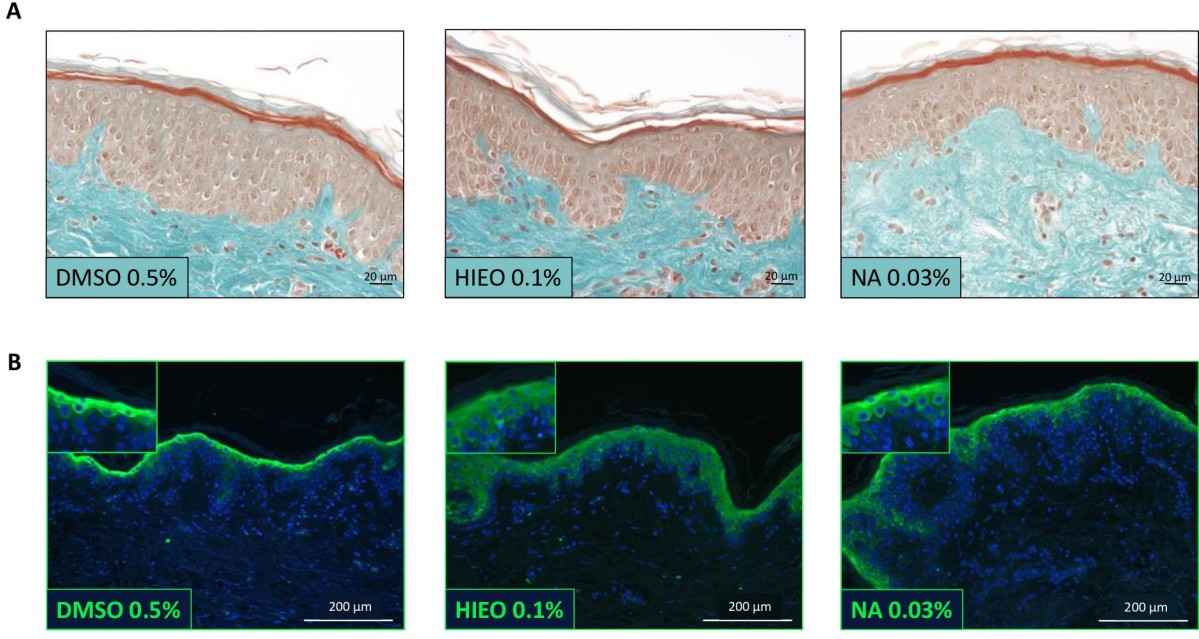

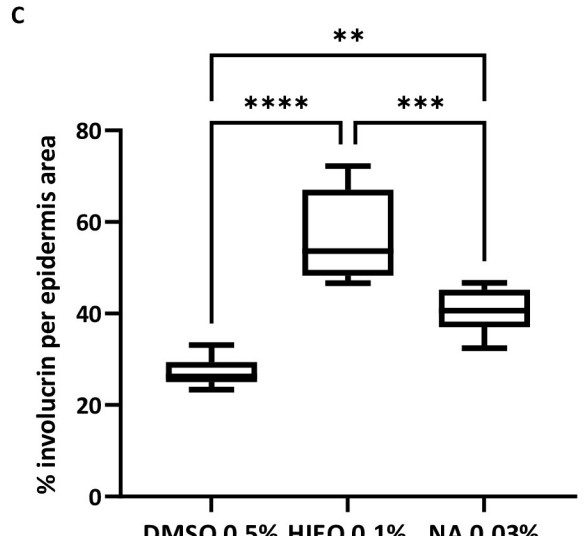

**Fig 6. NA increased IVL protein. A:** Histological and morphological analysis at day 5 using Masson's trichrome staining after application of DMSO 0.5%, HIEO 0.1% or NA 0.03%. Scale bar, 20 μm. **B:** Confocal microscopy observation of IVL immunostaining. Scale bare, 200 μm. IVL in green, nuclei in blue. **C:** Quantification of IVL surface area by image analysis. Data are expressed in % IVL per epidermis area. IVL data are presented as box and whisker plot where boxes span first and third quartiles, bars as the median values, and whiskers as the minimum and maximum mean of five representative images per replicate (n = 4). **p<0.01; ***p<0.001; ****p<0.0001 (Mann–Whitney test).

significant increase in IVL area related to the epidermis area for both HIEO and NA treatment. HIEO led to a more pronounced increase in expressions of IVL protein (+111 ± 37%) than NA (+51 ± 18%) compared to the control DMSO 0.5%.

## NA increases lipids and ceramides production

As show in Fig 7A, genes involved in the formation of TAG, linoleic acid, and the metabolism of cholesterol were regulated after 24 hours of NA and HIEO treatment. Indeed, *glycerol kinase (GK)* convert glycerol into glycerol 3 phosphate, the substrate of *glycerol-3-phosphate acyl-transferase 3 (GPAT3)* that catalyze the first step of TAG synthesis and *patatin like phospholipase domain containing 1 (PNPLA1)* in association with his cofactor *abhydrolase domain containing 5 (ABHD5)* is facilitate TAG lipolysis to produce linoleic acid. Among the various candidates that could hydrolyze TAG, the expression of *lipase family N* (LIPN) was increased by NA and HIEO treatment. *Fatty acid desaturase 6 (FADS6)*, coding for a bifunctional enzyme desaturating linoleic acid or α-linolenic acid was decreased by NA and HIEO

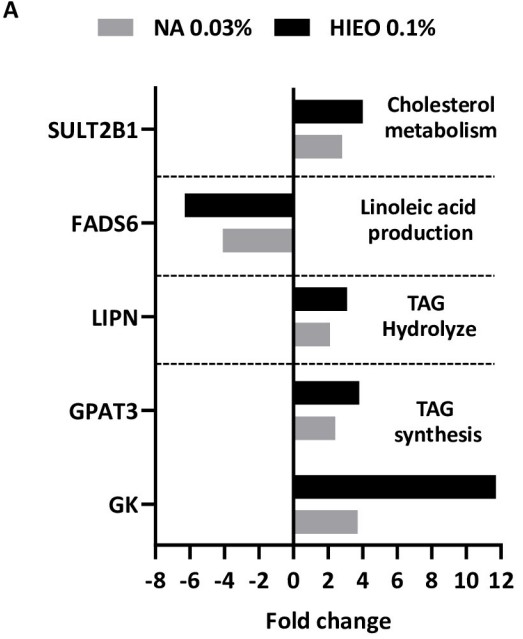

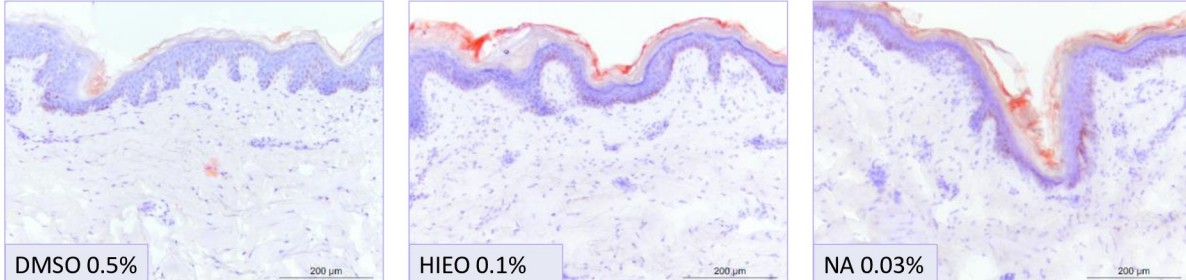

**Fig 7. NA increased lipids in the epidermis.** A: Transcriptome analysis was performed with microarray using skin explant treated with NA (0.03%) or HIEO (0.1%) and a control group, DMSO (0.5%). Mean fold-change induced by HIEO and NA in quadruplicate explant for each donor from three distinct donors after 24-hour treatment are shown. The 1008 genes in common to HIEO and NA were analyzed with IPA software and an enrichment analysis of certain pathways allowed the identification of differentially expressed genes (fold changes ≥ 2 or ≤ -2, FDR p-value < 0.05) involves TAG synthesis, linoleic acid synthesis and cholesterol metabolism. Mean fold-changes induced by HIEO and NA (NA) in quadruplicate explant for each donor from three distinct donors after 24-hour treatment are shown. B: Skin sections were stained with Oil Red O to detect neutral lipid in the epidermis at day 5 after DMSO 0.5%, NA 0.03% or HIEO 0.1% treatment. Optical representative images show the location of dye in the intercellular spaces of the epidermis. Lipids are stained in red. Scale bar, 200 μm.

treatment. In addition, increase of *perilipin 2* (*PLIN2)* was observed and it is responsible for the formation and accumulation of lipid storage droplet. We also observed an increase of expression of *phospholipase A2 group IVD* (*PLA2G4D)* that are critical for the conversion of phospholipid to fatty acid (FA) and *SULT2B1* related to cholesterol sulfate synthesis from cholesterol was raised with both treatments.

A general examination of lipid production using Oil Red O was performed (Fig 7B). This staining indicates that NA and HIEO increased the incorporation of lipids into the intercellular space of the epidermis compared to the control DMSO 0.5%. As shown in Fig 7B, HIEO treatment appears to increase more neutral lipid production than NA.

As show in Fig 8, genes involved in all steps of the acylglycosylceramide biosynthesis were induced after 24 hours of NA and HIEO treatment (Fig 8). Indeed, the expression of *ELOVL1*, *ELOVL4* and *ELOVL7* genes, involved in the elongation of fatty acids from palmitoyl-CoA in ultra-long-chain fatty acids (ULCFA), were increased by both treatments. *Cytochrome P450 family 4 subfamily F member 22 (CYP4F22)*, responsible for ω-hydroxylation of ultra-long-chain fatty acids (ULCFA) and acyl-CoA synthetase (*acyl-CoA synthase long chain family member 3, ACSL3*), ensuring ω-hydroxy-ULCFA conversion to ω-hydroxy-ULCFA-CoA, were both upregulated by NA and HIEO. *CERS3*, which catalyzes ceramide formation from sphinganine and acyl-CoA to produce ω-OH-ULC-FAs and *UGCG*, a membrane glucosyltransferase involved in the glycosylation of ceramides (up to 26 carbon atoms), were both upregulated.

To be secreted into the extracellular space at the interface between the SG and SC shortly before the keratinocytes transform into corneocytes, glucosylceramides (GLcCers) are packed into lamellar bodies by *ABCC12*. *ABCC12* gene expression was increased by NA and HIEO treatment as well as *TGM1*, which attaches GlcCers to the CE protein IVL.

Using LC/MS, we measured CER[NS] and CER[AS] in the epidermis of skin explants (n = 3) from three female donors (age range 36 to 49 years) treated with DMSO (0.5%) as the control, NA (0.03%) or HIEO (0.1%) for five days. As shown in Fig 9, the increase in total ceramides was significantly higher in the presence of HIEO (0.1%) than NA (0.03%), ranging from +114.2% to +64.2% compared to DMSO 0.5%.

6 Ceramide [NS] and 2 ceramide [AS] were identified and revealed that the treatment with NA or HIEO elicited a significant increase in almost all ceramides detected except CER[N(24:1)S(18)], which is decreased by both treatments. A significant difference was found for CER[N(16)S(18)], CER[N(18)S(18)] and CER[(14:1)S(18)] between HIEO and NA treatment. No difference was observed for CER[A(16)S(18)], CER[N(20)S(18)], CER[A(24)S(18)], CER [N(24)S(18)] and CER[N(26)S(18)] with either treatment.

As shown in Figs 8 and 9, differences between NA and HIEO were observed in neutral lipid staining and ceramide production. Indeed, transcriptomic analysis shown that only HIEO enhanced gene expression levels of *patatin like phospholipase domain containing 1* (*PNPLA1)* related to glycerophospholipid metabolism and decreased the expression of *PNPLA3* and *PNPLA4*, coding for lipases involved in TAG hydrolysis or TAG homeostasis. In addition, *serine palmitoyltransferase long chain base subunit 2* (*SPTLC2*) and *serine palmitoyltransferase small subunit B* (*SPTSSB*) encoding to key enzymes in sphingolipid biosynthesis and *Ceramide synthase 6* (*CerS6*), a key enzyme in the biosynthesis of short acyl chain ceramides, C14–C16 ceramides were only increased by HIEO treatment. By decreasing *alkaline ceramidase 1* (*ACER1*), HIEO treatment not favorizes the hydrolysis of very long chain ceramides to generate sphingosine (S3 Fig).

## Discussion

The chemical composition of the tested HIEO in our study is characteristic of Corsican HIEO, with NA (32.80%) as the major compound and a low concentration in α-pinene (2.02%) [10].

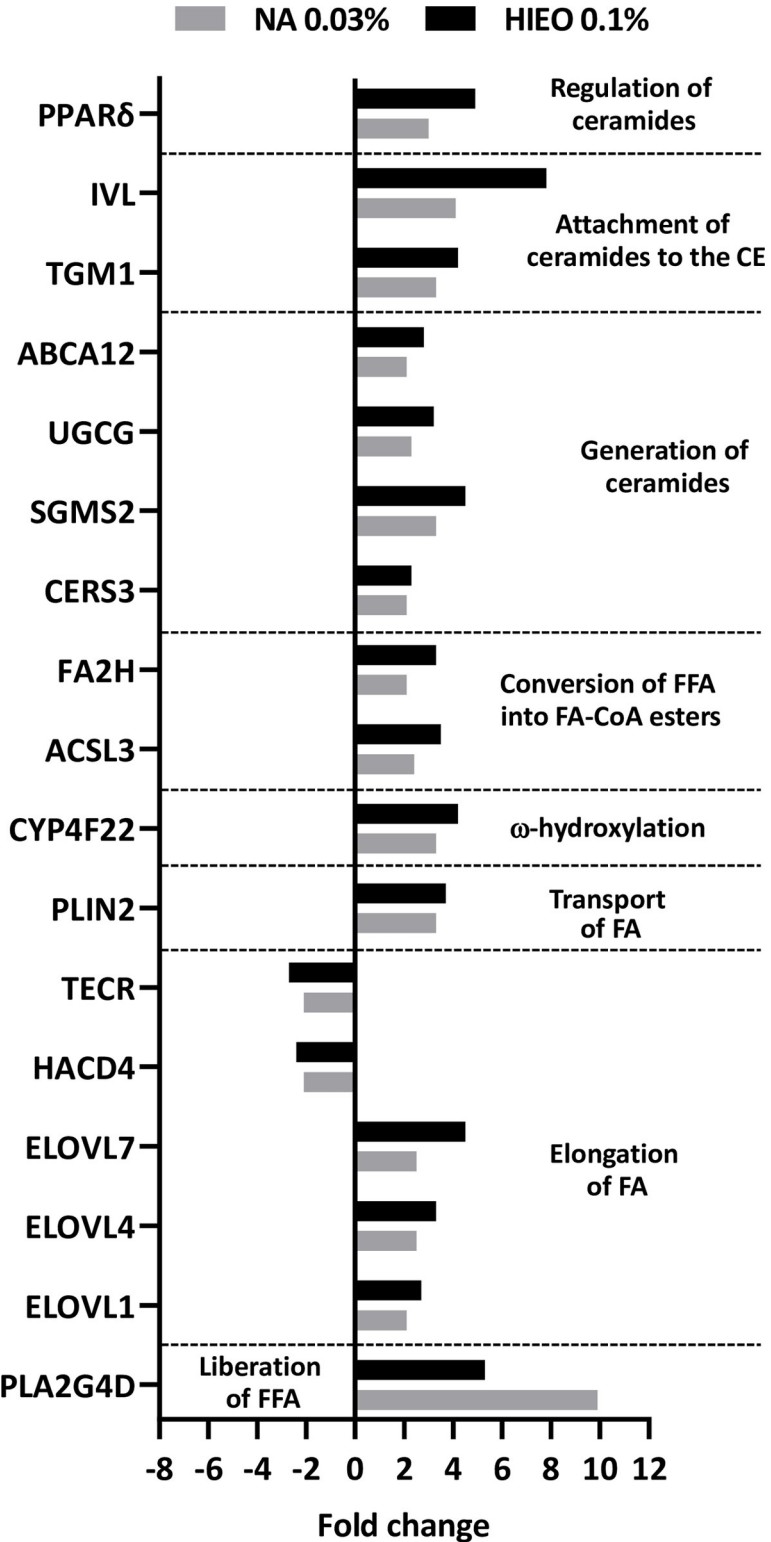

**Fig 8. NA increased the gene expression of ceramide synthesis-related enzymes.** Transcriptome analysis was performed with microarray using skin explant treated with NA (0.03%) or HIEO (0.1%) and a control group, DMSO (0.5%). Mean fold-change induced by HIEO and NA in quadruplicate explant for each donor from three distinct donors after 24-hour treatment are shown. The 1008 genes in common to HIEO and NA were analyzed with IPA software and an enrichment analysis of certain pathways allowed the identification of differentially expressed genes

(fold changes ≥ 2 or ≤ -2, FDR p-value < 0.05) involved in fatty acid (FA) elongation, ω-hydroxyceramide production and ceramide generation were analyzed. Mean fold-changes induced by HIEO and NA (NA) in quadruplicate explant for each donor from three distinct donors after 24-hour treatment are shown.

We previously demonstrated that Corsican HIEO effectively promotes skin barrier function and hydration [26].

Because NA is the major compound of HIEO from Corsica with unknown skin activity, we wanted to elucidate its role in the biological functions of HIEO. For that, we compared their biological activity by transcriptomic analysis; NA was tested at a dose equivalent to that found in Corsican HIEO. We discovered that HIEO and NA share 1,008 common genes and 41.5% of HIEO-regulated transcripts were due to NA, with genes involved in skin barrier formation, keratinocyte differentiation and ceramide synthesis.

The predominant problem with aging skin is a disturbed barrier function characterized by dryness, reduced immune response and slow wound healing [27]. The skin's primary function is to act as a protective barrier between the organism and its external environment, minimizing water loss from the body whilst preventing the entry of pathogens and allergens. Epidermal barrier dysfunction can exacerbate sensitive skin conditions, dryness and infections.

The analysis of regulated gene expression suggests that NA drives at least and HIEO have a significant effect on the keratinocyte differentiation process and promote skin barrier formation. Indeed, genes that code for protein-regulating differentiation are upregulated, with increased IVL expression at mRNA and protein level. This protein is incorporated as a component of the cornified envelope [28] via the TGM1 action that is also increased by NA and

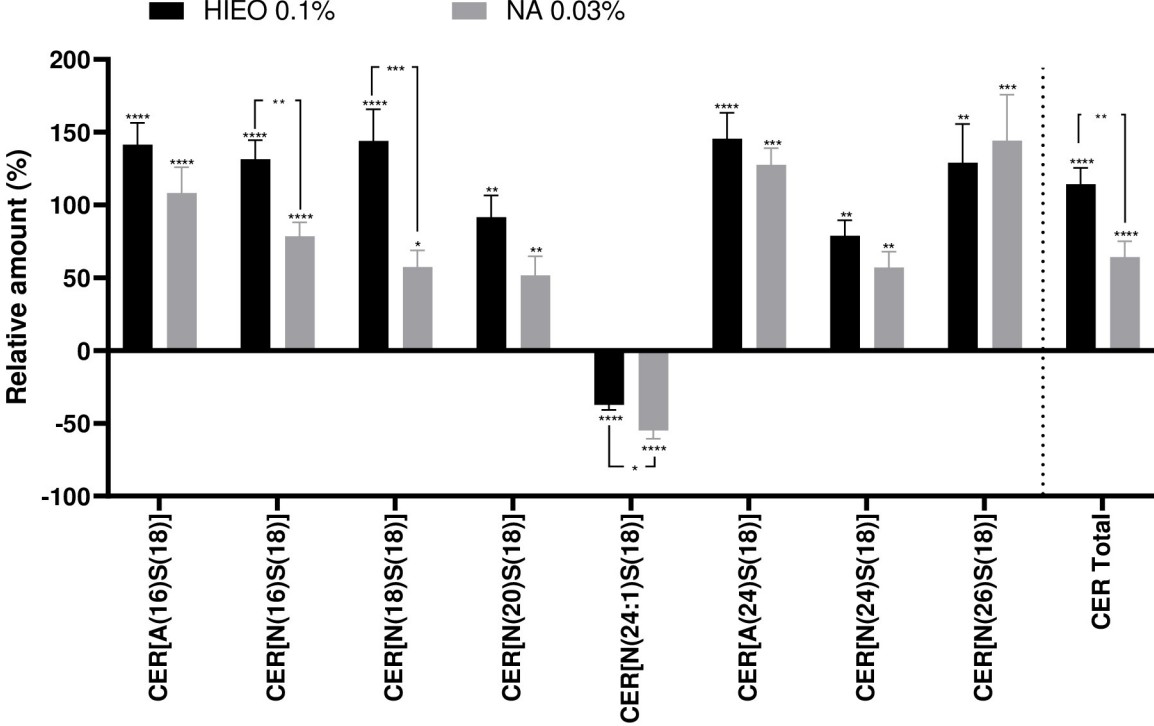

**Fig 9. NA increased Ceramides in the epidermis.** Ceramides were extracted from epidermis and subjected to LC/MS for determining the levels of each ceramide species. The amount of ceramide species in the HIEO (0.1%) or NA (0.03%) conditions is expressed as a percentage in comparison to the DMSO (0.5%) control condition, which was arbitrarily set (0%). The bar graph representing the relative amount of ceramides is set in %. *p<0.05; **p<0.01; ***p<0.001; ****p<0.0001 (One-way Anova with post-hoc Tukey HSD test).

HIEO treatment. By increasing IVL and *TGM1*, NA and HIEO contribute to barrier maintenance, and with the upregulation of *AQP3*, NA and HIEO should facilitate osmotically driven transport of the water and glycerol. *CRNN*, *S100A16*, *S100P* and *SPRR3* genes coding to cornified envelope proteins [29] were also upregulated with transcripts (*PRSS3*, *KLK13*) [30,31] involved in desquamation. Other differentiation regulators were upregulated, such as *GRHL3*, *CBS/CBSL*, *EPHA2*, *CTSL*, *PPARF064* and *RRAS2* [32]. Inversely, NA and HIEO decreased the expression of transcripts associated with keratinocyte proliferation, such as *VAV3*, *WNT16*, *FGFR2/3*, *KRT5* and *KRT14* [33,34], and may accelerate the differentiation that slows in the elderly population.

The increase in genes involved in junctions (*PKP2*, *DSCG3*, *CLDN7*, *GJB3*, *GJB5*) [35] suggests that NA and HIEO increase keratinocyte cohesion in addition to the development of cornified envelopes. Four keratinocyte growth factors, *EPGN*, *EREG*, *AREG* and *HB-EGF*, which can bind to *FGFR2/3* and regulate keratinocyte proliferation and/or migration [36,37], were upregulated by NA and HIEO. Therefore, FGFR2/3 downregulation is the likely mechanism to stop the action of growth factors [38]. We hypothesize that NA and HIEO firstly increase epidermal differentiation, cell cohesion and cornified envelope formation, and secondly induce *HB-EGF*, *EREG*, *AREG* and *EPGN* to facilitate wound healing in the epidermis, which is impaired in aged skin with delayed barrier recovery after injury.

Lipids are important constituents of the human epidermis. Either free and organized into broad lipid bilayers in the intercorneocyte spaces, or covalently bound to the corneocyte envelope, they play a crucial role in permeability barrier function as a blocker of transepidermal water loss. In aged skin, biochemical analysis of epidermal lipids reveals that the content of three major lipid species—ceramide, cholesterol and fatty acids—was reduced [25]. This could be explained by the reduced activity of each lipid's rate-limiting enzymes. By increasing lipid production, the application of HIEO or NA could reinforce barrier function, increase water retention and lower penetration of exogenous compounds. Cholesterol sulfate was not analyzed in our study but could also be increased by HIEO and NA treatment because the cholesterol sulfotransferase (*SULT2B1*) gene related to its synthesis [22] rises with both treatments.

Ceramides are sphingolipids that consist of a long chain of sphingosine base linked to long-chain fatty acid via an amide bond. In this study, we focused on the more common 18-carbon sphingosine base, the most abundant sphingoid base in the epidermis and known to promote keratinocyte differentiation [24] and non-hydroxylated (N) and hydroxylated (A) medium-chain ceramides that represent around 21 species. NA and HIEO increased the production of non-hydroxylated CER[N(20)S(18)], CER[N(24)S(18) and CER[N(26)S(18)] or hydroxylated CER[A(24)S(18)] at similar levels. This production can be attributed to the increasing expression of ELOVL1 and ELOVL4 [39] induced by HIEO and NA treatment. Ceramides with small-chain non-hydroxy and hydroxy fatty acid (16–18 carbons) were more strongly increased by HIEO than NA, probably correlated to the expression of ELOVL7 (C16 to C20-CoAs) [40] also rising more with HIEO than NA. Acyl-ceramides (EOS, EOH, EOP and EODS), which account for 12% of *stratum corneum* ceramides, were not evaluated by LC/MS analysis in this study. However, protein transcripts involved in the four steps of acyl-ceramide production were upregulated by NA and HIEO. In addition, PPAR$\delta$ transcript was upregulated by NA and HIEO, and is directly connected to ceramide production by inducing *ABCA12* expression [41].

We observed that NA and HIEO enhanced *PPARF064* expression, epidermis differentiation, and lipid and ceramide production. Since *PPAR$\delta$* stimulates keratinocyte differentiation and ceramide synthesis [42], it seems reasonable to consider the possible activation of *PPARF064* by NA content in HIEO, probably indirectly by the well-known PPAR$\delta$ ligand linoleic acid.

From the transcriptomic analysis carried out, it appears that the action of HIEO by its multiple composition seems to induce a differentiation of keratinocytes and a synthesis of ceramides which differs in part from NA and which would be investigated in another study.

In conclusion, our results demonstrate that NA, the major component of HIEO, strengthens the skin barrier function by increasing lipid and ceramide content in the SC through enhancing the expressions of ceramide synthesis-related enzymes required for the glucosylceramide pathway. Furthermore, both compounds increase the epidermal differentiation complex by stimulating the expression of IVL (transcript and protein). In addition, we demonstrate for the first time that HIEO-regulated effects in this study are mediated by its principal component, NA.

Therefore, we anticipate that NA, the major component of Corsican HIEO, may be effective in improving skin barrier function and moisture retention in age-associated skin conditions.

## Supporting information

**S1 Fig. HIEO increases the expression of genes involved in keratinocytes differentiation.** Transcriptome analysis was performed with microarray using skin explant treated with NA (0.03%) or HIEO (0.1%) and a control group, DMSO (0.5%). Mean fold-change induced by HIEO and NA in quadruplicate explant for each donor from three distinct donors after 24-hour treatment are shown. The 2,429-genes and 1,137-genes related to HIEO (0.1%) and to NA (0.03%) were analyzed with IPA software and an enrichment analysis of certain pathways allowed the identification of differentially expressed genes (fold changes $\geq 2$ or $\leq -2$, FDR p-value $< 0.05$) involved in the differentiation and proliferation of keratinocytes. (DOCX)

**S2 Fig. HIEO increases the expression of genes involved in epidermis structure.** Transcriptome analysis was performed with microarray using skin explant treated with NA (0.03%) or HIEO (0.1%) and a control group, DMSO (0.5%). Mean fold-change induced by HIEO and NA in quadruplicate explant for each donor from three distinct donors after 24-hour treatment are shown. The 2,429-genes and 1,137-genes related to HIEO (0.1%) and to NA (0.03%) were analyzed with IPA software and an enrichment analysis of certain pathways allowed the identification of differentially expressed genes (fold changes $\geq 2$ or $\leq -2$, FDR p-value $< 0.05$) involved in the structure of the epidermis. (DOCX)

**S3 Fig. HIEO increased the gene expression of ceramide synthesis-related enzymes.** Transcriptome analysis was performed with microarray using skin explant treated with NA (0.03%) or HIEO (0.1%) and a control group, DMSO (0.5%). Mean fold-change induced by HIEO and NA in quadruplicate explant for each donor from three distinct donors after 24-hour treatment are shown. The 2,429-genes and 1,137-genes related to HIEO (0.1%) and to NA (0.03%) were analyzed with IPA software and an enrichment analysis of certain pathways allowed the identification of differentially expressed genes (fold changes $\geq 2$ or $\leq -2$, FDR p-value $< 0.05$) involved in fatty acid (FA) elongation, ω-hydroxyceramide production and ceramide generation were analyzed. Mean fold-changes induced by HIEO and NA (NA) in quadruplicate explant for each donor from three distinct donors after 24-hour treatment are shown. (DOCX)

**S1 Table. List of the 1,008 commun genes between NA 0.03% and HIEO 0.1%.** (XLSX)

**S2 Table. List of the 129 genes only regulated by NA 0.03%.**
(XLSX)

**S3 Table. List of the 1,421 genes only regulated by HIEO 0.1%.**
(XLSX)

**S4 Table. List of 59 enriched GO terms for biological processes, molecular functions, and cellular components related to 1,008 genes common to NA and HIEO.** Corrected p-value: Since multiple GO accession number are tested for their significance, hence a multiple testing correction is performed. The GO spreadsheet is sorted based on the corrected p-values. Count in Selection: This refers to the number of genes in the selected entity (for example, from T-test) list which have that particular GO term or its descendants. Count in Total: This refers to the number of genes in All Entities which have that GO term or its descendants. % Count in Selection: This refers to the percentage of genes in the input entity list which have that GO term.
(XLSX)

**S1 Data.**
(7Z)

## Author Contributions

**Conceptualization:** Géraldine Lemaire, Valérie Cenizo.

**Data curation:** Géraldine Lemaire, Malvina Olivero, Virginie Rouquet, Alain Moga, Aurélie Pagnon.

**Formal analysis:** Géraldine Lemaire, Malvina Olivero, Virginie Rouquet, Alain Moga, Aurélie Pagnon.

**Methodology:** Géraldine Lemaire, Alain Moga, Aurélie Pagnon.

**Supervision:** Géraldine Lemaire, Valérie Cenizo, Pascal Portes.

**Validation:** Géraldine Lemaire.

**Writing – original draft:** Géraldine Lemaire.

**Writing – review & editing:** Géraldine Lemaire, Virginie Rouquet, Alain Moga, Aurélie Pagnon, Valérie Cenizo, Pascal Portes.

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
