## [Decision Letter · Decision Letter 0]

6 Sep 2022

PONE-D-22-12384Neryl acetate, the major component of Corsican Helichrysum Italicum essential oil, mediates its biological activities on skin barrierPLOS ONE

Dear Dr Lemaire.

Thank you for submitting your manuscript to PLOS. I would also like to thank the reviewers for the time they spent on this work. After careful consideration, we feel that it has merit but does not fully meet PLOS ONE’s publication criteria as it currently stands

You will find that several points need to be addressed to improve the quality of this work. In particular, you are encouraged to better exploit your data and to correct inaccuracies and errors regarding epidermal differentiation and lipid metabolism.

My decision is, therefore "major revision" and we invite you to submit a revised version of the manuscript that addresses the points raised during the review process

Dr. Thierry Chardot

We look forward to receiving your revised manuscript.

Kind regards,

Thierry Chardot, PhD

Academic Editor

PLOS ONE

“Laboratoires M&L SA - Groupe L'Occitane funded the study. Co-authors Géraldine Lemaire, Malvina Olivero, Virginie Rouquet, Valérie Cenizo and Pascal Portes are employed by Laboratoires M&L SA - Groupe L'Occitane. Laboratoires M&L SA - Groupe L'Occitane provided support in the form of salaries for authors GL, MO, VR, VC, PP. Co-author Alain Moga is employed by QIMA - Synelvia. QIMA – Synelvia provided support in the form of salaries for author AM. Co-author Aurélie Pagnon is employed by Novotec.  Novotec provided support in the form of salaries for author AP. G.L: supervise research, performed research, analyzed data, and wrote the paper, M.O: performed research and analyzed data, V.R: performed research, analyzed data and review the paper, A.P: performed research, analyzed data and review the paper, A.M: performed research, analyzed data and review the paper, V.C: supervise research and review the paper, P.P: supervise research and review the paper. The financial support from Laboratoires M&L - Groupe L'Occitane does not alter the authors’ adherence to all PLOS ONE policies on sharing data and materials and there are no restrictions on sharing of data and/or materials.”

Reviewers' comments:

Reviewer's Responses to Questions

**Comments to the Author**

1. Is the manuscript technically sound, and do the data support the conclusions?

Reviewer #1: Yes

Reviewer #2: Partly

Reviewer #3: Yes

2. Has the statistical analysis been performed appropriately and rigorously? 

Reviewer #1: Yes

Reviewer #2: Yes

Reviewer #3: Yes

3. Have the authors made all data underlying the findings in their manuscript fully available?

Reviewer #1: Yes

Reviewer #2: Yes

Reviewer #3: Yes

4. Is the manuscript presented in an intelligible fashion and written in standard English?

Reviewer #1: Yes

Reviewer #2: Yes

Reviewer #3: Yes

5. Review Comments to the Author

Reviewer #1: Manuscript Number: PONE-D-22-12384 

In the manuscript “Neryl acetate, the major component of Corsican Helichrysum Italicum essential oil, mediates its biological activities on skin barrier“ the authors analyzed the biological regulations in the skin explant by transcriptomic analysis, skin barrier protein immunofluorescence, lipid staining and ceramide analysis by liquid chromatography-mass spectrometry. To publication in plosone some major points in the manuscript must be clarified, as follow

1. the name of the plants' genus must be abbreviated in the text after the first appearance.

2. In the Material and Methods, the authors must indicate the voucher specie

3. Check the whole paper to remove spelling mistakes specially the species names and insert blanks where necessary.

4. In abstract, “Helichrysum Italicum essential oil” appear many times, please abbreviate it

5. Introduction, no italicized “Asteraceae”

6. in material and methods, how the authors identified the chemical composition without GC-MS analysis, GC-FID given only the % of each peak?

7. Table1, inform the calculate and literature retention index, split the 6 and 13 compound peaks to inform the % of each compound

Reviewer #2: In this manuscript, the authors investigated the biologic effect of neryl acetate (NA), the major component of a plant essential oil, on the epidermal differentiation of skin explant models. By mean of transcriptomic analyses (microarrays, quantitative RT-PCR), histological staining (immunofluorescence, lipid labelling) and ceramide quantification (LC-MS/MS), they concluded that NA modulated the expression of genes involved in epidermal differentiation, skin barrier formation and ceramide synthesis. Hence, NA could be an interesting molecule to improve skin-barrier function and moisture retention in aged skin.

Transcriptomic and lipidomic approaches provide an in-depth analysis of NA and HIEO biological effect. However, the analysis of the data and the structure of the manuscript, especially the results section, have to be improved. The study was focused on genes that are similarly regulated by NA and HIEO. It would be interesting to have information about the genes specifically regulated by the essential oil or NA alone, as each of these ingredients could be of interest for skin treatment.

Major comments

- The corneocytes result from the terminal differentiation of epidermal keratinocytes. As it, they do not have a “cytoplasm” (as written for instance p.4 line57, 64 and 70) but and intracellular matrix essentially made of intermediated filaments aggregates. Similarly, CE formation provides a mechanic resistance to the corneocyte rather than “cellular cohesion” (p.4 line 61), which results from SC intercellular junctions. This has to be corrected.

- The high level of NA in HIEO was already reported (p.3 line 52). Thus, the first paragraph in the Results section (p.12) about HIEO composition could be presented as supporting information or in materials and methods.

- The subsections in the Results section are very short and look like a succession of experiments and figures. The authors could regroup some of them, especially “NA increases the formation of the skin barrier” and “NA increases involucrin protein”, as well as the 3 subsections about epidermal lipids (p.16-17).

- The transcriptomic analysis has been performed on skin explants treated with HIEO and NA, but also with the vehicle alone (DMSO 0,5%), this has to be specified (p.13 line 243).

- In the transcriptomic analysis, the authors report the number of “probe sets”, “regulated transcripts” or “genes”, this is confusing. In the table S1, there are 1322 probes including 129 with no corresponding genes, furthermore, several probes may correspond to the same gene. Could the authors clarify?

- There are a lot of genes which are regulated by HIEO only (1865, cf Fig1). The authors focused their study on genes similarly regulated by NA and HIEO, but it would be interesting to know if some of these 1865 genes are also involved in the regulation of epidermal differentiation, barrier formation and ceramide synthesis, or in other cellular functions. The same question could be asked for the 155 genes specifically regulated by NA, especially if NA could be used as a cosmetic ingredient.

- There are around 30 genes represented in the Fig.2. Are there only 30 genes among the 1322 differentially regulated genes which are involved in barrier function, keratinocyte differentiation and ceramide synthesis, and which meet the criteria of an activation z-score of 2.3? In the legend of Fig2, the authors have to specify what are the different geometric boxes (diamond, square, oval, triangle, circle).

- I don’t agree with the comment of the authors concerning the regulation of the KRT genes. During epidermal differentiation, the expression of suprabasal keratins (K1, K10, K2) is induced while the expression of basal keratins (K5, K14) is downregulated. Concomitantly, there is an induction of the expression of numerous differentiation markers (IVL, LOR, TGM1…). Thus, the authors cannot explain that the downregulation of all the KRT genes reported in Fig4B can be explained by the induction of AQP3, TGM1, IVL and other late epidermis differentiation genes. Moreover, KRT2 is presented in Fig4B as a gene expressed in the basal layer, while it is on the contrary a late differentiation gene expressed only in the granular layer of the epidermis, later that KRT1 and KRT10.

- I was surprised that the authors present GK, GPAT3 and LIPN as genes involved in the acylglucosylceramide formation (p.16 line 325). Moreover, could the author explain how these genes are involved in the formation of linoleic acid (p.16 line327)?? To my knowledge, GK and GPAT3 are involved in the synthesis of triglyceride and glycerophospholipids. Concerning LIPN, its function in the metabolism of epidermal lipids is not yet clearly known. Those genes cannot be reported as “involved in acylglucosylceramide formation”, as well as some other genes reported in Fig6 like SULT2B1, involved in the metabolism of cholesterol.

- The Oil Red O staining of skin implant sections provides information on the presence of neutral triglycerides and lipids in the SC (Fig7). I would rather have presented this figure earlier, in order to introduce the analysis of the expression of genes involved in SC lipid metabolism and the quantification of epidermal ceramides.

- The Fig8 and its legend need improvements. I understand that the amount of ceramide species in the HIEO/NA conditions is expressed as a percentage in comparison to the DMSO control condition, which was arbitrarily set (0%). This could be clearly explained in the legend and specified “relative amount (%)” on the y-axis of the figure.

Minor comments:

- Gene names are generally written in italics.

- Ceramides constitute a hydrophobic extracellular matrix, not an « hydrophilic extracellular matrix » as written p.5 line 91.

- All along the manuscript, the authors should be more specific in the phrasing of some sentences. For instance, p.6 line 110, p.14 line 282 or p.15 line 304, they should specify that it is “the expression of” genes which is correlated with, or increases.

- Globally, legends of the figures are very short, they could be more detailed.

Reviewer #3: The paper evaluated the neryl acetate at a concentration similar to that found at Corsican Helichrysum Italicum essential oil as a mediator of biological activity on human skin. The paper is very well written and the experiment is very well designed. Few reviews can be done, as shown below.

Minor comment

1. The introduction needs to be explained, in one sentence, how the practical importance of skin protein expression, and how is the importance of the treatment that interferes with these expressions.

2. Line 30 – the family name didn’t need to be in italic

3. Line 44, 115 – the scientific name, at the second time of citation should be H. italicum italicum

4. In all ul or ml the letter l should be in capital letter, please review the entire paper

6. PLOS authors have the option to publish the peer review history of their article (what does this mean?). If published, this will include your full peer review and any attached files.

Reviewer #1: **Yes: **Pablo Luis Baia Figueiredo

Reviewer #2: No

Reviewer #3: **Yes: **Livio Martins Cos Junior

---

## [Author Response · Author response to Decision Letter 0]

4 Nov 2022

Dear Editor,

Thank you for your considerations on my manuscript and thanks to the reviewers for the time they spent to my works. 

Please find my responds to each point raised by the academic editor and reviewer 

The manuscript was corrected to meet PLOS ONE’s style requirements 

There is no grant number, the work was supported by Laboratoires M&L SA - Groupe L'Occitane and I’m an employee of Laboratoires M&L SA - Groupe L'Occitane. The funding information was corrected below.

Funding: This work was supported by Laboratoires M&L SA - Groupe L'Occitane. The funder provided support in the form of salaries for authors [G.L, M.O, V.R, V.C, P.P] and research materials, but did not have any additional role in study design, data collection and analysis, decision to publish, or preparation of the manuscript. A.M is employed by QIMA – Synelvia and A.P is employed by Novotec. The specific roles of each author are articulated in the ‘author contributions’ section.

Competing interests: The financial support from Laboratoires M&L - Groupe L'Occitane does not alter the authors’ adherence to all PLOS ONE policies on sharing data and materials and there are no restrictions on sharing of data and/or materials.

Reviewer #1: Manuscript Number: PONE-D-22-12384 

In the manuscript “Neryl acetate, the major component of Corsican Helichrysum Italicum essential oil, mediates its biological activities on skin barrier “the authors analyzed the biological regulations in the skin explant by transcriptomic analysis, skin barrier protein immunofluorescence, lipid staining and ceramide analysis by liquid chromatography-mass spectrometry. To publication in plosone some major points in the manuscript must be clarified, as follow.

1. the name of the plants' genus must be abbreviated in the text after the first appearance.

The name of plants’ genus was abbreviated in the text after the first appearance as you request 

2. In the Material and Methods, the authors must indicate the voucher specie

I do not understand what you mean by "the voucher specie", could you clarify?

3. Check the whole paper to remove spelling mistakes specially the species names and insert blanks where necessary.

The paper was checked to remove spelling mistakes and corrections were made.

4. In abstract, “Helichrysum Italicum essential oil” appear many times, please abbreviate it

Helichrysum italicum essential oil was abbreviated 

5. Introduction, no italicized “Asteraceae”

Asteraceae was not italicized

6. in material and methods, how the authors identified the chemical composition without GC-MS analysis, GC-FID given only the % of each peak?

Gas chromatography analysis was performed on a gas chromatograph equipped with a mass spectrometry detector, using a TR-WaxMS-fused silica capillary column (Thermo Fisher Scientific, 60 m x 0.25 mm i.d.; film thickness 0.25 μm). Chromatographic conditions were as follows: hydrogen as carrier gas at 0.7 mL/min and injector and detector temperatures at 250°C each. Oven temperature was isothermal at 60°C for 1 min, then increased to 240°C at a rate of 2°C/min and held isothermal for 23 mins. The volume injected was 1 μl with a split ratio of 200:1. Identification was performed with standards and/or NIST database.

7. Table1, inform the calculate and literature retention index, split the 6 and 13 compound peaks to inform the % of each compound

In table 1, 6 and 13 compounds cannot be separated because chromatogram present a coelution. The % of each will be not representative and underestimate.

Retention index is not used in this study, only retention time and comparison with standards is used. GC-MS analysis before GC-FID analysis confirm the compound identification 

Reviewer #2: In this manuscript, the authors investigated the biologic effect of neryl acetate (NA), the major component of a plant essential oil, on the epidermal differentiation of skin explant models. By mean of transcriptomic analyses (microarrays, quantitative RT-PCR), histological staining (immunofluorescence, lipid labelling) and ceramide quantification (LC-MS/MS), they concluded that NA modulated the expression of genes involved in epidermal differentiation, skin barrier formation and ceramide synthesis. Hence, NA could be an interesting molecule to improve skin-barrier function and moisture retention in aged skin. Transcriptomic and lipidomic approaches provide an in-depth analysis of NA and HIEO biological effect. However, the analysis of the data and the structure of the manuscript, especially the results section, have to be improved. 

The study was focused on genes that are similarly regulated by NA and HIEO. It would be interesting to have information about the genes specifically regulated by the essential oil or NA alone, as each of these ingredients could be of interest for skin treatment.

Major comments

- The corneocytes result from the terminal differentiation of epidermal keratinocytes. As it, they do not have a “cytoplasm” (as written for instance p.4 line57, 64 and 70) but and intracellular matrix essentially made of intermediated filaments aggregates. Similarly, CE formation provides a mechanic resistance to the corneocyte rather than “cellular cohesion” (p.4 line 61), which results from SC intercellular junctions. This has to be corrected.

This was corrected as you request

- The high level of NA in HIEO was already reported (p.3 line 52). Thus, the first paragraph in the Results section (p.12) about HIEO composition could be presented as supporting information or in materials and methods.

This first paragraph is now presented in materials and methods

- The subsections in the Results section are very short and look like a succession of experiments and figures. The authors could regroup some of them, especially “NA increases the formation of the skin barrier” and “NA increases involucrin protein”, as well as the 3 subsections about epidermal lipids (p.16-17).

The 3 subsections were regrouped in one section and idem for epidermal lipids

- The transcriptomic analysis has been performed on skin explants treated with HIEO and NA, but also with the vehicle alone (DMSO 0,5%), this has to be specified (p.13 line 243).

Vehicle alone (DMSO 0.5%) was specified (p13 line 243) as suggested and, also p13 line 247 for clarification.

- In the transcriptomic analysis, the authors report the number of “probe sets”, “regulated transcripts” or “genes”, this is confusing. In the table S1, there are 1322 probes including 129 with no corresponding genes, furthermore, several probes may correspond to the same gene. Could the authors clarify?

The text was corrected to clarify it and only the term gene was used in the text. A new table S1 has been added with the identified genes only and redundant probe sets or probe sets without corresponding genes have been excluded.

There are a lot of genes which are regulated by HIEO only (1865, cf Fig1). The authors focused their study on genes similarly regulated by NA and HIEO, but it would be interesting to know if some of these 1865 genes are also involved in the regulation of epidermal differentiation, barrier formation and ceramide synthesis, or in other cellular functions. The same question could be asked for the 155 genes specifically regulated by NA, especially if NA could be used as a cosmetic ingredient.

Genes specifically regulated by essential oil or NA alone were added in Table S3 and Table S2, respectively. An additional Figure 2 with GOs and Table S4 with the set of 59 GOs corresponding to the genes in common were added.

The analysis of HIEO genes in its entirety is not the focus of this paper and will be the subject of another paper, as we decided to focus on Neryl acetate which regulates about 40% of the HIEO genes. However, some genes that may explain the differences in expression of ceramides and neutral lipids have been added to the results with an additional table and a paragraph in the discussion has been added to highlight the differences. 

- There are around 30 genes represented in the Fig.2. Are there only 30 genes among the 1322 differentially regulated genes which are involved in barrier function, keratinocyte differentiation and ceramide synthesis, and which meet the criteria of an activation z-score of 2.3? 

The IPA software identified only 30 genes among the 1,008 genes in common to determine the 2.3 activation z-score criteria for skin differentiation. Other genes were identified in the list of 1008 as being involved in barrier function, keratinocyte differentiation, and ceramide synthesis using the scientific literature and were added in the manuscript.

In the legend of Fig2, the authors have to specify what are the different geometric boxes (diamond, square, oval, triangle, circle).

A legend has been added to specify the different geometric boxes

- I don’t agree with the comment of the authors concerning the regulation of the KRT genes. During epidermal differentiation, the expression of suprabasal keratins (K1, K10, K2) is induced while the expression of basal keratins (K5, K14) is downregulated. Concomitantly, there is an induction of the expression of numerous differentiation markers (IVL, LOR, TGM1…). Thus, the authors cannot explain that the downregulation of all the KRT genes reported in Fig4B can be explained by the induction of AQP3, TGM1, IVL and other late epidermis differentiation genes. 

This sentence was deleted 

Moreover, KRT2 is presented in Fig4B as a gene expressed in the basal layer, while it is on the contrary a late differentiation gene expressed only in the granular layer of the epidermis, later that KRT1 and KRT10. 

This was corrected in the Fig 4B.

- I was surprised that the authors present GK, GPAT3 and LIPN as genes involved in the acylglucosylceramide formation (p.16 line 325). Moreover, could the author explain how these genes are involved in the formation of linoleic acid (p.16 line327)?? To my knowledge, GK and GPAT3 are involved in the synthesis of triglyceride and glycerophospholipids. Concerning LIPN, its function in the metabolism of epidermal lipids is not yet clearly known. Those genes cannot be reported as “involved in acylglucosylceramide formation”, as well as some other genes reported in Fig6 like SULT2B1, involved in the metabolism of cholesterol.

This was corrected in the manuscript and the genes involved in TAG synthesis were put with Oil Red O stained skin implant sections and the function of LIPN in epidermal lipid metabolism was indicated as a hypothesis and the function of SULT2B1 was corrected

- The Oil Red O staining of skin implant sections provides information on the presence of neutral triglycerides and lipids in the SC (Fig7). I would rather have presented this figure earlier, in order to introduce the analysis of the expression of genes involved in SC lipid metabolism and the quantification of epidermal ceramides.

The figure has been moved to be presented before with the genes involved in triglyceride and lipid synthesis in the SC. 

- The Fig8 and its legend need improvements. I understand that the amount of ceramide species in the HIEO/NA conditions is expressed as a percentage in comparison to the DMSO control condition, which was arbitrarily set (0%). This could be clearly explained in the legend and specified “relative amount (%)” on the y-axis of the figure.

This was corrected.

Minor comments:

- Gene names are generally written in italics. 

Genes names were written in italics

- Ceramides constitute a hydrophobic extracellular matrix, not an « hydrophilic extracellular matrix » as written p.5 line 91.

This was corrected.

- All along the manuscript, the authors should be more specific in the phrasing of some sentences. For instance, p.6 line 110, p.14 line 282 or p.15 line 304, they should specify that it is “the expression of” genes which is correlated with, or increases.

This was corrected as suggested

- Globally, legends of the figures are very short, they could be more detailed.

Legends were more detailed as suggested

 

Reviewer #3: The paper evaluated the neryl acetate at a concentration similar to that found at Corsican Helichrysum Italicum essential oil as a mediator of biological activity on human skin. The paper is very well written and the experiment is very well designed. Few reviews can be done, as shown below.

Minor comment

1. The introduction needs to be explained, in one sentence, how the practical importance of skin protein expression, and how is the importance of the treatment that interferes with these expressions.

2. Line 30 – the family name didn’t need to be in italic 

This was corrected

3. Line 44, 115 – the scientific name, at the second time of citation should be H. italicum italicum

This was corrected 

4. In all ul or ml the letter l should be in capital letter, please review the entire paper

This was corrected

---

## [Decision Letter · Decision Letter 1]

6 Jan 2023

PONE-D-22-12384R1Neryl acetate, the major component of Corsican Helichrysum Italicum essential oil, mediates its biological activities on skin barrierPLOS ONE

Dear Dr. Lemaire

Thank you for submitting your revised version of the manuscript to PLOS ONE. You have been taking in account the reviewers critiques, however there are still few points that should be adresse before acceptance.Therefore, we invite you to submit a revised version of the manuscript that addresses the points raised during the review process.

Kind regards,

Thierry Chardot, PhD

Academic Editor

PLOS ONE

Journal Requirements:

Reviewers' comments:

Reviewer's Responses to Questions

**Comments to the Author**

1. If the authors have adequately addressed your comments raised in a previous round of review and you feel that this manuscript is now acceptable for publication, you may indicate that here to bypass the “Comments to the Author” section, enter your conflict of interest statement in the “Confidential to Editor” section, and submit your "Accept" recommendation.

Reviewer #2: (No Response)

2. Is the manuscript technically sound, and do the data support the conclusions?

Reviewer #2: Yes

3. Has the statistical analysis been performed appropriately and rigorously? 

Reviewer #2: Yes

4. Have the authors made all data underlying the findings in their manuscript fully available?

Reviewer #2: Yes

5. Is the manuscript presented in an intelligible fashion and written in standard English?

Reviewer #2: Yes

6. Review Comments to the Author

Reviewer #2: The authors significantly improved the manuscript and took into account the remarks.

There are still minor corrections to do:

- In the introduction, please take into account the difference between cornified envelope (CE) and corneocyte: line 62, about the major components of the stratum corneum, it would be more correct to write “four major components: the corneocyte, surrounded by the cornified envelope (CE) and the cornified lipid envelope (CLE), which are embedded in the intercellular lipid layers”. Similarly, line 68, write “fill the intracellular matrix of corneocytes” (instead of CE). Finally, line 74, write “the main reinforcement protein for the CE on its intracellular face” (instead of cytoplasmic face)

- In the Materials and Methods section, lane 217: Table 2 instead of Table 1.

- In the Results section, line318: KRT2 is downregulated as KRT1 and KRT10, thus correct the text.

- In Fig5C: Please change “Tight junctions” by “intercellular junctions” (which include other junctions than TJs)

- In the Results section, line 357: It’s better writing “involved in the formation of TAG and linoleic acid, and the metabolism of cholesterol”. Line 378: “involves TAG synthesis, linoleic acid synthesis and cholesterol metabolism”.

- There is a confusion between LIPN and PNPLA1. Line 360 (result section) and lines 495-496 (Discussion section): this is PNPLA1 and not LIPN which works in association with ABHD5 to synthesize omega-O-acylceramides. This has to be corrected. As I previously notified concerning LIPN, its function in the metabolism of epidermal lipids is not yet clearly known.

7. PLOS authors have the option to publish the peer review history of their article (what does this mean?). If published, this will include your full peer review and any attached files.

Reviewer #2: No

---

## [Author Response · Author response to Decision Letter 1]

17 Feb 2023

PONE-D-22-12384R1

Neryl acetate, the major component of Corsican Helichrysum Italicum essential oil, mediates its biological activities on skin barrier

PLOS ONE

Dear Editor,

Please find my answer to the few points resting. 

Kind regards,

Géraldine Lemaire

Reviewer #2: The authors significantly improved the manuscript and took into account the remarks.

There are still minor corrections to do:

- In the introduction, please take into account the difference between cornified envelope (CE) and corneocyte: line 62, about the major components of the stratum corneum, it would be more correct to write “four major components: the corneocyte, surrounded by the cornified envelope (CE) and the cornified lipid envelope (CLE), which are embedded in the intercellular lipid layers”. Similarly, line 68, write “fill the intracellular matrix of corneocytes” (instead of CE). Finally, line 74, write “the main reinforcement protein for the CE on its intracellular face” (instead of cytoplasmic face)

The sentences were corrected

- In the Materials and Methods section, lane 217: Table 2 instead of Table 1.

This was corrected 

- In the Results section, line318: KRT2 is downregulated as KRT1 and KRT10, thus correct the text.

This was corrected

- In Fig5C: Please change “Tight junctions” by “intercellular junctions” (which include other junctions than TJs)

This was corrected in the figure 5C

- In the Results section, line 357: It’s better writing “involved in the formation of TAG and linoleic acid, and the metabolism of cholesterol”. Line 378: “involves TAG synthesis, linoleic acid synthesis and cholesterol metabolism”.

This was corrected 

- There is a confusion between LIPN and PNPLA1. Line 360 (result section) and lines 495-496 (Discussion section): this is PNPLA1 and not LIPN which works in association with ABHD5 to synthesize omega-O-acylceramides. This has to be corrected. As I previously notified concerning LIPN, its function in the metabolism of epidermal lipids is not yet clearly known.

This was corrected as the following sentences: As show in Fig 7A, genes involved in the formation of TAG, linoleic acid, and the metabolism of cholesterol were regulated after 24 hours of NA and HIEO treatment. Indeed, glycerol kinase (GK) convert glycerol into glycerol 3 phosphate, the substrate of glycerol-3-phosphate acyltransferase 3 (GPAT3) that catalyze the first step of TAG synthesis and patatin like phospholipase domain containing 1 (PNPLA1) in association with his cofactor abhydrolase domain containing 5 (ABHD5) is facilitate TAG lipolysis to produce linoleic acid. Among the various candidates that could hydrolyze TAG, the expression of lipase family N (LIPN) was increased by NA and HIEO.

And the sentence was deleted in the discussion as following: Since PPARδ stimulates keratinocyte differentiation and ceramide synthesis [42], it seems reasonable to consider the possible activation of PPAR� by NA content in HIEO, probably indirectly by the well-known PPAR� ligand linoleic acid.

---

## [Editor Report · Decision Letter 2]

21 Feb 2023

Neryl acetate, the major component of Corsican Helichrysum Italicum essential oil, mediates its biological activities on skin barrier

PONE-D-22-12384R2

Dear Dr. lemaire

We’re pleased to inform you that your manuscript has been judged scientifically suitable for publication and will be formally accepted for publication once it meets all outstanding technical requirements.

Kind regards,

Thierry Chardot, PhD

Academic Editor

PLOS ONE

Additional Editor Comments:

Thank you for your patience and for all the positive work improvements.
---

## [Editor Report · Acceptance letter]

23 Feb 2023

PONE-D-22-12384R2 

Neryl acetate, the major component of Corsican *Helichrysum italicum* essential oil, mediates its biological activities on skin barrier 

Dear Dr. Lemaire:

I'm pleased to inform you that your manuscript has been deemed suitable for publication in PLOS ONE. Congratulations! Your manuscript is now with our production department. 

Kind regards, 

on behalf of

Dr Thierry Chardot 

Academic Editor

PLOS ONE